# Observation of topological transport quantization by dissipation in fast Thouless pumps

Zlata Fedorova [1✉], Haixin Qiu [1✉], Stefan Linden [1✉] & Johann Kroha [1✉]

Quantized dynamics is essential for natural processes and technological applications alike. The work of Thouless on quantized particle transport in slowly varying potentials (Thouless pumping) has played a key role in understanding that such quantization may be caused not only by discrete eigenvalues of a quantum system, but also by invariants associated with the nontrivial topology of the Hamiltonian parameter space. Since its discovery, quantized Thouless pumping has been believed to be restricted to the limit of slow driving, a fundamental obstacle for experimental applications. Here, we introduce non-Hermitian Floquet engineering as a new concept to overcome this problem. We predict that a topological band structure and associated quantized transport can be restored at driving frequencies as large as the system's band gap. The underlying mechanism is suppression of non-adiabatic transitions by tailored, time-periodic dissipation. We confirm the theoretical predictions by experiments on topological transport quantization in plasmonic waveguide arrays.

[1] Physikalisches Institut, Rheinische Friedrich-Wilhelms-Universität Bonn, Nussallee 12, 53115 Bonn, Germany. ✉email: cherpakova@physik.uni-bonn.de; haixin@th.physik.uni-bonn.de; linden@physik.uni-bonn.de; kroha@th.physik.uni-bonn.de

The standard realization for Thouless pumping[1,2] is the time-periodic version of the Rice-Mele (RM) model[3], which describes a dimerized tight-binding chain whose system parameters change cyclically along a closed loop in Hamiltonian parameter space. In the adiabatic regime and for a completely filled band, the net particle transfer per cycle is an integer given alone by the Berry phase associated with the loop or the Chern number of the band, i.e., a topological invariant robust against topology-preserving deformations of the parametric loop. Such nontrivial topology of the Hamiltonian parameter space or band structure was recognized as the overarching concept behind phenomena apparently as diverse as the integer quantum Hall effect[4], the quantum spin Hall effect[5], topological insulators in solid state[6] and photonics[7,8], quantum spin[9] or charge pumping[1], Dirac or Weyl semimetals[10], and the electric polarization of crystalline solids[11]. Recently, topological or Thouless pumping was experimentally demonstrated using ultracold atoms in dynamically controlled optical lattices[12,13] or using waveguide arrays[14].

In realistic systems, however, Thouless pumping generically faces two difficulties. First, at nonzero driving frequencies, unavoidable in experiments, the system becomes topologically trivial. The reason is that the nonzero driving frequency defines a Floquet-Bloch Brillouin zone (FBBZ) and the dimension of the band structure is increased by one compared to the adiabatic case. The coupling between forward-propagating and backward-propagating states then opens a gap[15–17], so that the Chern number, or winding number around the FBBZ, of the effectively two-dimensional band becomes trivial, and the particle transport deviates from perfect quantization. Second, realistic experimental systems are to some extent open and subject to dissipation, so that the quantum mechanical time evolution of single-particle states deviates from unitarity, which may prevent the closing of the cycle in Hamiltonian parameter space. This motivates the interest in non-Hermitian (NH) Hamiltonians. Non-Hermiticity can have profound influence on the system dynamics. In addition to ubiquitous exponential decay, it may cause such peculiar phenomena as dissipation-induced localization in the Caldeira-Legget model[18], unidirectional robust transport[19], asymmetric transmission or reflection[20,21], or NH topological edge states associated with exceptional points[22–24]. Non-Hermiticity has been utilized to probe topological quantities[25,26]. Another fascinating example is the so-called non-Hermitian shortcut to adiabaticity[27–29], which describes faster evolution of a wavefunction in an NH system than in its Hermitian counterpart.

Here, we introduce time-periodic modulation of dissipation as a new concept to restore topological transport quantization in fast Thouless pumps. Although in many-body systems dissipation would be induced by interactions or particle loss, the plasmon polariton dynamics in our experiments is mathematically identical to that of a linear, dissipative, periodically driven Schrödinger equation. To analyze systems of this kind theoretically, we utilize the Floquet theory for non-Hermitian, time-periodic systems. Using this formalism, we demonstrate for a driven RM model that time-periodic dissipation can give rise to a band structure in the two-dimensional FBBZ with a nontrivial Chern number. Hence, the mean displacement of a wave packet per cycle is quantized even when the driving frequency is fast, i.e., far from adiabaticity. In a real-space picture, this topologically quantized transport comes about, because the time-periodic loss selectively suppresses the hybridization of a right(left)-moving mode with the counterpropagating one. The theoretical predictions are confirmed by experiments on arrays of coupled dielectric-loaded surface plasmon-polariton waveguides (DLSPPW)[30]. DLSPPWs are uniquely suited model systems for realizing topological transport with dissipation: The propagation of surface plasmon polaritons mathematically realizes the single-particle Schrödinger equation on a one-dimensional tight-binding lattice[30,31], where the waveguide axis resembles time, and the system parameters, including losses, can easily be modulated along the waveguide axis. Moreover, complete band filling is achieved via Fourier transform to k-space by pumping a single site (waveguide) of the tight-binding lattice. This is essential for probing the band topology which otherwise is possible only in fermionic systems at low temperature.

## Results

**Model.** We consider a periodically driven RM model[17,32] with additional onsite, periodic dissipation (see Fig. 1), $\hat{H}(t) = \hat{H}_{RM}(t) - i\hat{\Gamma}(t)$,

$$\hat{H}_{RM}(t) = \sum_j \left( J_1(t)\hat{b}_j^\dagger\hat{a}_j + J_2(t)\hat{a}_{j+1}^\dagger\hat{b}_j + h.c. \right)$$
$$+ \sum_j \left( u_a(t)\hat{a}_j^\dagger\hat{a}_j + u_b(t)\hat{b}_j^\dagger\hat{b}_j \right),$$
(1)

$$\hat{\Gamma}(t) = \sum_j \left( \gamma_a(t)\hat{a}_j^\dagger\hat{a}_j + \gamma_b(t)\hat{b}_j^\dagger\hat{b}_j \right).$$
(2)

where $j$ runs over all unit cells, $\hat{H}_{RM}(t)$ is the Hamiltonian of the periodically driven, nondissipative RM model and $\hat{\Gamma}(t)$ describes the losses. $\hat{a}_j^\dagger$ and $\hat{b}_j^\dagger$ ($\hat{a}_j$ and $\hat{b}_j$) are creation (annihilation) operators in unit cell $j$ on sublattice $A$ and $B$, respectively. The inter-/intra-cell hopping amplitudes, $J_{1/2}(t)$ and the onsite potentials on the two sublattices, $u_a(t)$ and $u_b(t)$, are all real-valued, periodic functions of time with frequency $\Omega = 2\pi/T$ according to

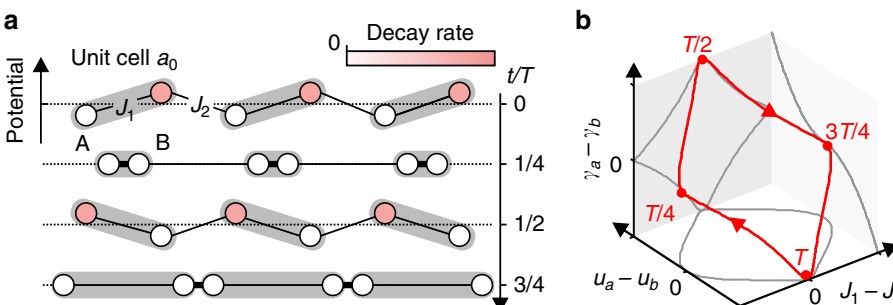

**Fig. 1 Non-Hermitian driven Rice-Mele model. a** Schematic of the periodically driven, NH RM lattice for four equidistant times during a pumping cycle. Lossy sites are depicted by red color, large (small) hopping amplitudes $J_{1,2}$ by short (long) distances between sites. **b** Pumping cycle in the parameter space $(J_1 - J_2, u_a - u_b, \gamma_a - \gamma_b)$.

$$u_a(t) = -u_0 \cos(\Omega t + \varphi), \quad u_b(t) = u_a(t - T/2),$$
$$J_1(t) = J_0 e^{-\lambda(1 - \sin \Omega t)}, \qquad J_2(t) = J_1(t - T/2),$$

with $u_0$, $J_0$, $\lambda > 0$, and $\varphi = 0$ (unless otherwise specified). The choice of the hopping amplitudes is motivated by the exponential dependence of the wave-function overlaps on the spacing $\lambda(1 - \sin \Omega t)$ between neighboring sites, as in our experiment below. In our NH modification of the RM model, the time-periodic decay rates $\gamma_a(t) \geq 0$ and $\gamma_b(t) \geq 0$ are nonzero once the onsite potential exceeds the mean value $[u_a(t) + u_b(t)]/2 = 0$. This resembles, for instance, a realistic situation where particles in a trapping potential are lost from the trap once the trapping potential is not sufficiently deep. Thus, we choose

$$\gamma_a(t) = -\gamma_0 \, \Theta(u_a(t)) \, \cos(\Omega t + \varphi), \quad \gamma_b(t) = \gamma_a(t - T/2),$$

where $\Theta(x)$ is the Heaviside step function.

**Non-Hermitian Floquet analysis.** In the following calculations we use the non-Hermitian Floquet formalism described in the "Methods" section below. We assume $u_0 = J_0 = 1$, $\lambda = 1.75$ and all energies are given in units of $J_0$.

In view of the experimental setup discussed below, we also consider the time evolution of states $|\Psi_A(t)\rangle$ and $|\Psi_B(t)\rangle$ which have been initialized ("injected") at time $t = 0$ with nonzero amplitude only at a single site of the $A$ or $B$ sublattice, respectively. For the parametric cycle configuration shown in Fig. 1b the chosen initial time moment leads to an asymmetric amplitude distribution of the counter-propagating Floquet states with respect to the two sublattices. As seen in Fig. 2, such initial conditions populate, by Fourier expansion, almost homogeneously an entire right-moving or left-moving band. Thus, it is a way to create the topologically important complete band filling, which would otherwise be possible only in fermionic systems. In the Hermitian case ($\gamma_0 = 0$), we see from Fig. 2a that the counterpropagating bands hybridize, accompanied by avoided crossings and gaps with width $G$ opening at the Floquet Brillouin zone boundaries, so that the bands become topologically trivial. As a result, the charge pumped per period deviates from the quantized value. This marks the generic breakdown of quantized Thouless pumping at any finite pumping frequency $\Omega$, as also noted in[15,17]. Note that computing the gap size $G$, as visible in Fig. 2a, involves diagonalization of the entire Floquet Hamiltonian matrix. In leading order perturbation theory, $G$ would be given by the Fourier amplitude of the periodic drive, i.e., for the first FBBZ by $J_0$, which strongly differs from the exact value.

We now consider the NH RM model driven with $\gamma_0 = 0.4 J_0$ (see Fig. 2b–e). Adding losses leads to several profound effects. First, the quasienergies become complex, whereby the right-moving and left-moving bands acquire considerably different dampings shown in Fig. 2e and seen as different broadenings of the spectral band occupation in Fig. 2b, c. Second, the two inputs are no longer equivalent in respect to the relative populations of the two bands. In particular, for the input $A$ we almost exclusively excite right-moving states, while for the input $B$ in addition to the lossy left-moving states, we partially populate right moving-states. Third, and most importantly, the gap $G$ closes and, hence, the bands wind around the entire 2D FBBZ as illustrated in Fig. 2d. In the "Methods" section it is shown that this restores the quantized transport (see Eq. (17)). Note, that these effects only occur once $\gamma_0$ is larger than some threshold value. In order to study this threshold behaviour we numerically evaluated the gap size $G$ at various driving frequencies and loss amplitudes (see Fig. 2f). In the Hermitian case ($\gamma_0 = 0$) the gap size has a complex oscillatory behaviour[33] as a function of the driving frequency.

Our analysis shows that a larger gap size requires stronger damping in order to close it. For instance, at the previously analyzed driving frequency $\Omega = 1.1 J_0$ the loss amplitude $\gamma_0$ should be larger than $0.3 J_0$ to close the gap.

Next, we investigate the position of the center of mass (CoM) of the wave-packet, $\langle x \rangle(t) = \langle \Psi(t)|x|\Psi(t)\rangle / \langle \Psi(t)|\Psi(t)\rangle$, after up to 5 completed driving cycles at various losses and fixed driving frequency $\Omega = 1.1 J_0$ for different initial conditions input $A$ or $B$ (see Fig. 3a, b). In the adiabatic case the mean displacement is almost $+1$ ($-1$) unit cell per cycle for delta-like excitations on sublattice $A$ ($B$). Small deviations from unity result from slight inhomogeneity of the band population. At the driving frequency $\Omega = 1.1 J_0$ the displacement per cycle is considerably smaller in the Hermitian case ($\gamma = 0$) indicating deviation from the quantized transport. With increasing losses this deviation becomes smaller and smaller for input $A$ and for $\gamma \geq 0.3$ the displacement can not be distinguished from the adiabatic case. Surprisingly, for the input $B$ we observe that the CoM position switches direction with time. This is a signature of the chirality of the Floquet bands and is due to the fact that the propagation of even poorly populated low-loss states in positive $x$-direction starts to dominate after the first few periods, while the states propagating in negative $x$-direction are quickly damped due to the phase relation of the periodic losses with respect to the hopping amplitude.

**Experiments.** In order to test our theoretical predictions we performed experiments based on DLSPPWs. The experimental realization of the model described by Eqs. (1) and (2) are based on the mathematical equivalence between the time-dependent Schrödinger equation in tight-binding approximation and the paraxial Helmholtz equation which describes propagation of light in coupled waveguide arrays[30,31]. Figure 4 shows a scheme of a DLSPPW array (a) as well as a scanning electron micrograph, (b) and an AFM scan, (c) of a typical sample. The sample fabrication process and the typical geometrical parameters of the arrays are described in the "Methods" section. The waveguide array represents a dimerized 1D lattice, where each unit cell contains two waveguides, $A$ and $B$. Here, the propagation direction $z$ plays the role of time. Periodic modulation of the effective hopping amplitudes is reached by sinusoidally varying the spacing between the adjacent waveguides $d_{1,2}(z)$ while the on-site potential variation is realized by changing the waveguides' cross-sections (heights $h_{a,b}(z)$ and widths $w_{a,b}(z)$). In addition, the variation of the waveguides' cross-section affects the instantaneous losses $\gamma_{a,b}(z)$. When the cross-section decreases, the confinement of the guided mode weakens. As a result, the modes can couple to free-propagating surface plasmon polaritons (SPPs) and scatter out from the array. We employ this effect to introduce time-dependent losses $\gamma_{a,b}(z)$.

We first consider a pumping cycle that encloses the critical point. For this purpose we choose the geometrical parameters of the DLSPPW array such that $u_0 = 1.1 J_0$ and $\Omega = 1.45 J_0$. By comparing the real-space intensity distribution to numerical calculations we estimate the loss amplitude to be $\gamma_0 = 0.8 J_0$. The real-space SPP intensity distribution $I(x, z)$ recorded by leakage radiation microscopy (see "Methods" section) for single site excitation at site A is shown in Fig. 5a. According to the aforementioned quantum optical analogy this corresponds to the probability density $I(x, t) = |\Psi(x, t)|^2$. We observe for all $z$ a strongly localized wave packet, whose CoM is transported in positive $x$-direction in a quantized manner, i.e., by one unit cell per driving period (see dotted lines), even though the driving frequency $\Omega$ is larger than the modulation amplitudes of all relevant parameters.

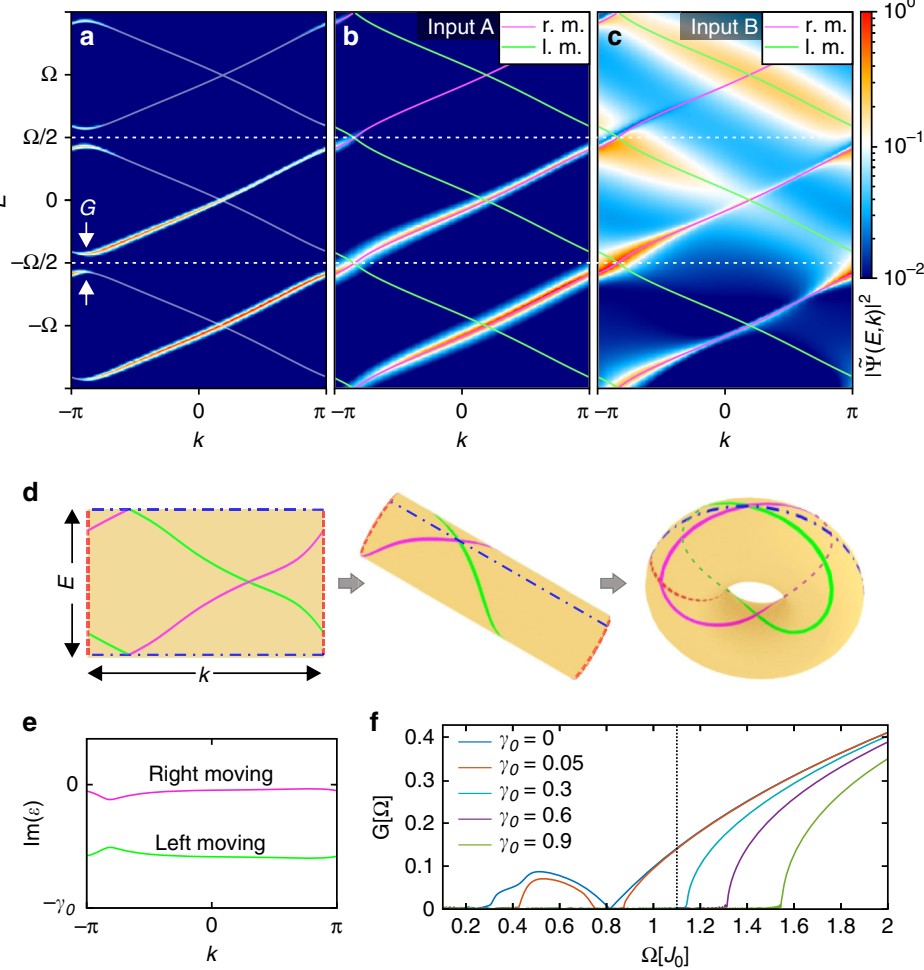

**Fig. 2 Floquet analysis of the driven NH RM model in the non-adiabatic regime.** Calculated band structures of the RM model for driving frequency $\Omega = 1.1J_0$. Thin lines: left-moving and right-moving Floquet quasienergy bands (real parts). **a** Band structure of the Hermitian RM model ($\gamma_0 = 0$). The band gaps at $E = \pm\Omega/2$ indicate a topologically trivial band structure, i.e., the breakdown of transport quantization. color code: normalized spectral occupation density of a state $|\Psi(t)\rangle$ injected at time $t = 0$ on a single site of the sublattice $A$, calculated from Eq. (13). It is seen that this injection almost homogeneously populates the right-moving bands, and almost no mixing of different Floquet modes occurs, as described by Eq. (14). **b** Same as in **a** but for the NH RM model with $\gamma_0 = 0.4J_0$ when the system is excited at a single site of the initially nonlossy sublattice $A$. As in **a**, almost no mixing of Floquet modes occurs. The gap at the FBBZ boundary is closed, restoring transport quantization. **c** Same as in **b**, but for a state injected at a site of the initially lossy sublattice $B$. Although the band gaps remain closed by the dissipation, this predominantly populates the left-moving band with a broad distribution, and the losses are high. **d** The first FBBZ which evolves into a 2D torus due to the periodicity along the $E$ axis (coincidence of dashed-dotted lines), as well as the $k$ axis (coinciding dashed lines). The magenta and green lines are the forward-propagating and backward-propagating dispersions analogous to **b** and **c**. They wind around the torus with winding numbers $Z = \pm1$ (c.f. "Methods" section). **e** Imaginary part of the quasienergy bands presented in **b**, **c**, showing low dissipation in the right-moving band. **f** The size of the band gap $G$ in dependence on the driving frequency at different loss amplitudes $\gamma_0$. The black dashed line shows $\Omega = 1.1J_0$.

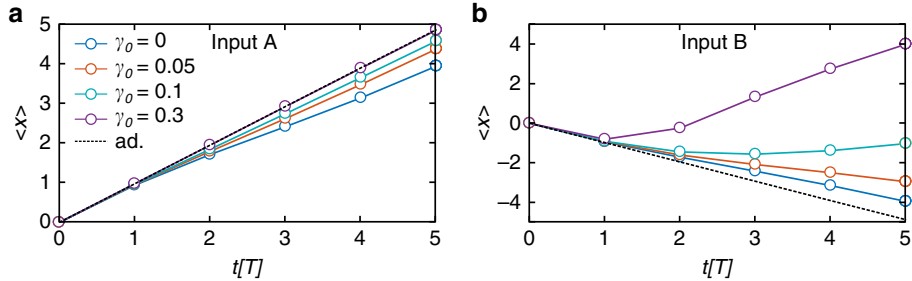

**Fig. 3 The center-of-mass shift in the NH RM model.** The center-of-mass position of the injected wavepacket after up to 5 full pumping cycles ($\Omega = 1.1J_0$) at different loss amplitudes $\gamma_0$ for a single-site input on (**a**) sublattice $A$ and (**b**) sublattice $B$.

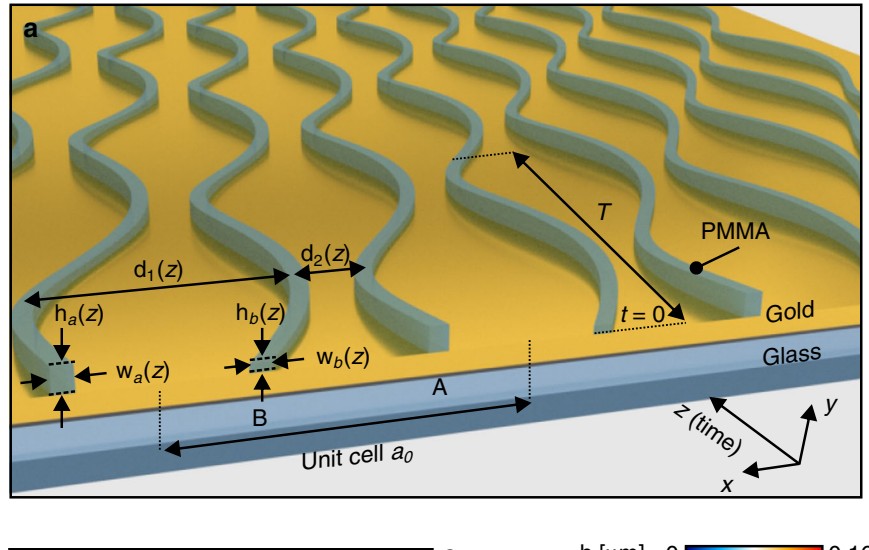

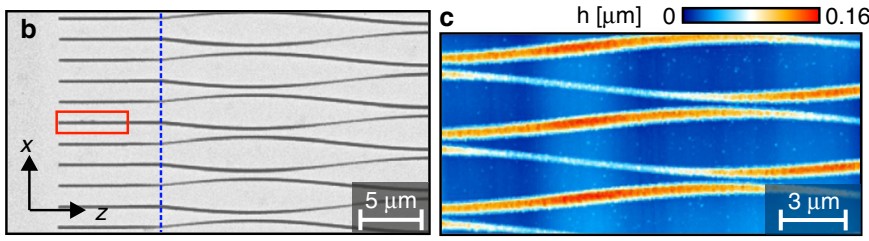

**Fig. 4 Plasmonic implementation of the NH RM model. a** Sketch of the plasmonic implementation of the NH RM model. **b** Scanning electron micrograph of a typical sample corresponding to $J_0 = 0.144\,\mu m^{-1}$, $\Omega = 1.45 J_0$, $u_0 = 1.1\,J_0$, $\gamma_0 = 0.8\,J_0$. The red dotted box highlights the grating coupler deposited onto the input waveguide A. **c** AFM scan of the same sample as shown in **b**.

The corresponding momentum resolved spectrum $I(k_x, k_z)$ is obtained by Fourier-space leakage radiation microscopy[34] and is shown in Fig. 5b. This intensity distribution is analogous to the spectral energy density presented in Fig. 2. We note that this technique provides the full decomposition in momentum components in the higher Brillouin zones[30]. The main feature of the spectrum is a continuous band with average slope $a_0/T$. The absence of gaps in the band indicates that the band winds around the 2D FBBZ $\{-\Omega/2 \le k_z < \Omega/2; -\pi/a_0 \le k_x < \pi/a_0\}$. This is a hallmark of a quantized pumping and confirms our theoretical predictions (see Fig. 2b).

As a reference measurement, we consider the parametric cycle, where all parameters are changing with the same amplitudes as in the previous case but the phase is chosen as $\varphi = \pi/2$. Under these conditions the Hamiltonian is symmetric under space and time inversion. In Fig. 5c we present the real-space SPP intensity distribution for this parametric cycle. In contrast to the previous case the wave packet is spreading and we do not observe CoM transport in $x$-direction. The corresponding momentum resolved spectrum shows a complicated band structure with multiple band gaps (see Fig. 5d). Obviously, none of the bands winds around the 2D FBBZ.

Directional transport of light in periodically curved waveguides can be in principle also achieved by using a simple combination of directional couplers with constant effective mode index, i.e., constant waveguide cross-section[35,36]. However, due to periodic exchange of power between two coupled waveguides this effect has a resonant character and the period of modulation plays in this case a crucial role. In order to demonstrate that the directional transport in our system has a different origin, we repeat the experiment shown in Fig. 5a for three different driving frequencies $\Omega$ (0.7$J_0$, 1.1$J_0$, 1.45$J_0$). Moreover we prepare two sets

of samples, one with modulation of the waveguide cross-section as before ($u_0 = 1.1J_0$, $\gamma_0 = 0.8J_0$) and the second with constant cross-section ($u_0 = 0$, $\gamma_0 = 0$). The measured real space intensity distributions are depicted in Fig. 6a. We extract from this data the CoM position after up to 4 complete periods as displayed in Fig. 6b. In the case with cross-section modulation (red markers) the CoM is shifted by one unit cell per period $T$ at all chosen driving frequencies. We note that the somewhat lower than unit slope of the CoM plots in Figs. 6b, 7b during the first pumping cycle is an artefact which arises from non-ideal excitation conditions, such as weak excitation of the neighbouring waveguides. The deviations at large distance are statistical and result from increasing measurement errors due to camera noise and decaying signal intensity. In Fourier space changing the modulation frequency influences primarily the width of the Floquet BZ: the lower is the frequency, the smaller is the distance between the neighbouring bands and the smaller is the tilt of these bands which reflects the wavepacket group velocity in absolute values (see Supplementary Fig. 1).

Without cross-section modulation (blue markers in Fig. 6b) the CoM displacement per period at these frequencies is much smaller than in the quantized case and depends on the driving frequency. These measurements confirm that the observed directional transport in our system is not a resonant directional coupler effect.

Up to now we only considered experiments with excitation at sub-lattice A (low loss input). The numerical calculations predict that the transport in the opposite direction for single site excitation at sub-lattice B is strongly suppressed by the time-periodic losses. To test this, we perform additional experiments to study how the transport properties depend on the initial conditions for different strengths of cross-section modulation.

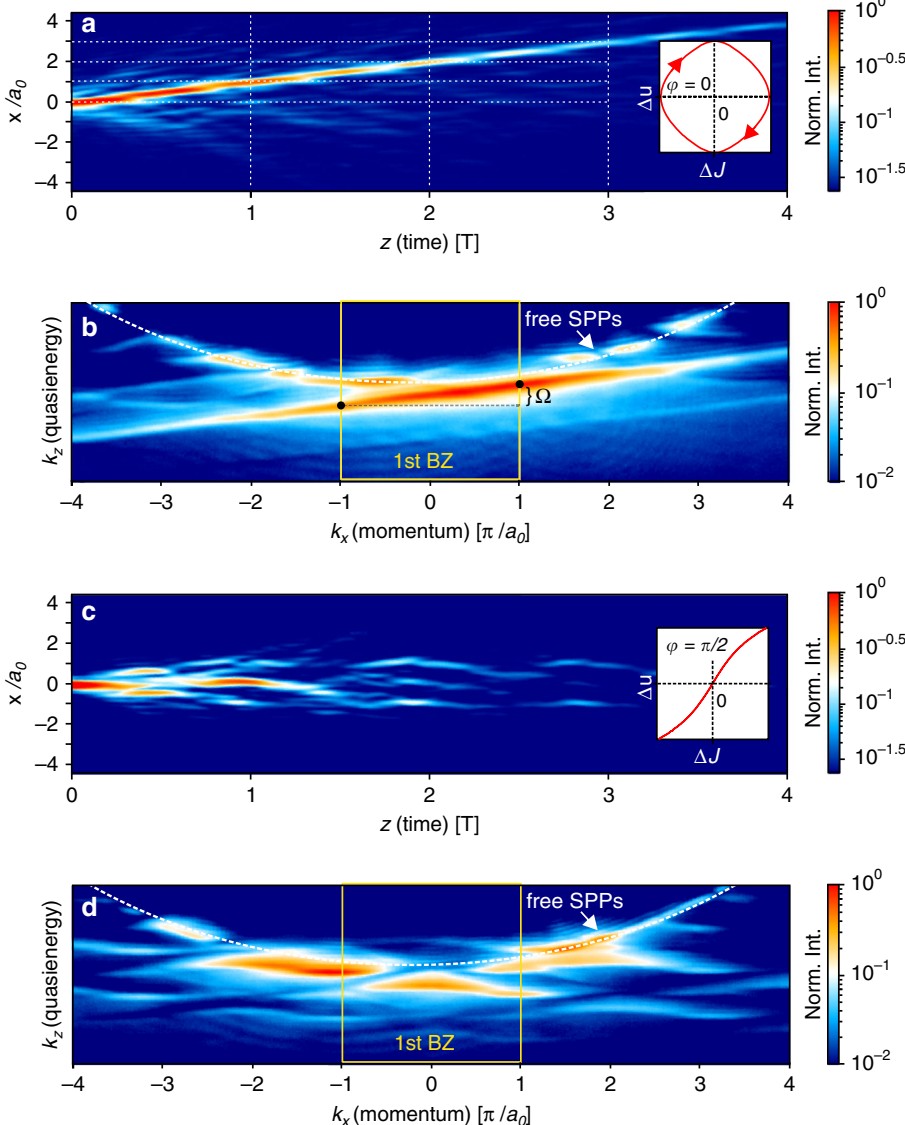

**Fig. 5 Observation of fast Thouless pumping in a DLSPPW array. a** Real-space SPP intensity distribution for $u_0 = 1.1J_0$, $\gamma_0 = 0.8J_0$, and $\varphi = 0$. Plot on the right shows the projection of the corresponding pumping cycle onto the plane $(J_1 - J_2, u_a - u_b)$. **b** Fourier-space SPP intensity distribution corresponding to **a**. **c** Real-space SPP intensity distribution for $u_0 = 1.1J_0$, $\gamma_0 = 0.8J_0$, and $\varphi = \pi/2$. Plot on the right again shows the corresponding cycle in parameter space. **d** Fourier-space SPP intensity distribution corresponding to **c**.

In doing so we tune the amplitude of the on-site potential $u_0$ and simultaneously the loss amplitude $\gamma_0$. Fig. 7a shows the real space intensity distributions for the excitation at the waveguides A (left column) and B (right column) for three different cross-section modulations and the driving frequency $\Omega = 1.45J_0$. The CoM displacement derived from this data is depicted in Fig. 7(b) (waveguide A: circles, waveguide B: triangles). In case of small modulation strength ($u_0 = 0.3J_0$, $\gamma_0 = 0.1J_0$, red markers) SPPs excited at site A and B are transported in $+x$ and $-x$ directions, respectively. However, for both inputs the mean displacement of the CoM is less than 1 unit cell per period. For the modulation strength ($u_0 = 1.1J_0$, $\gamma_0 = 0.8J_0$, blue markers) input A shows quantized displacement of the CoM while the sign of the mean displacement for input B switches from $+$ to $-$. This effect becomes even stronger at higher modulation strength $u_0 = 1.5J_0$, $\gamma_0 = 1.1J_0$ (green)—as predicted by theory (compare with Fig. 3a–b). In Fourier space increasing the modulation strength results in a strong band broadening caused by a growing damping

rate. This effect is more pronounced for the input B (see Supplementary Fig. 2).

## Discussion

In this work, we introduced the concept of time-periodic dissipation in Floquet topological systems. The theoretical analysis required the generalization of Floquet theory to quantum mechanics with non-Hermitian, time periodic Hamiltonians. Such quantum systems can be simulated experimentally in dielectric-loaded surface-plasmon polariton waveguide (DLSPPW) arrays. Specifically, we considered a non-Hermitian extension of the periodically driven Rice-Mele model. While fast driving of dissipationless systems always destructs the quantization of Thouless pumping, we predicted theoretically that time-periodic and space-periodic dissipation can lead to the restoration of quantized transport for nonadiabatic driving conditions. This finding results from the fact that periodic loss can modify the

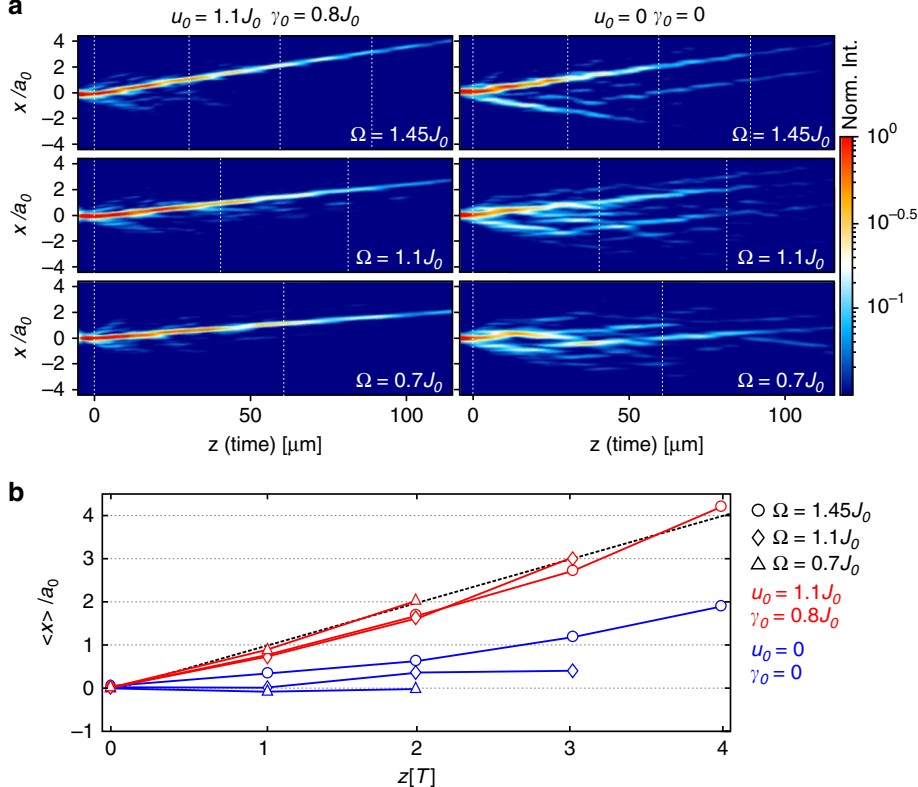

**Fig. 6 Influence of driving frequency on the transport. a** Real-space SPP intensity distributions for different driving frequencies and single-site excitation at waveguide A. The left and right column correspond to arrays with cross-section modulation ($u_0 = 1.1J_0$, $\gamma_0 = 0.8$) and without cross-section modulation ($u_0 = 0$, $\gamma_0 = 0$), respectively. **b** The CoM position of the SPP intensity in dependence on propagation distance $z$ calculated from the experimental results shown in **a**. Note that the $z$-axis here is normalized to the period $T$. Red markers correspond to arrays with cross-section modulation and blue markers correspond to no modulation. The black dashed line shows the anticipated adiabatic behaviour.

Floquet-Bloch band structure in such a way that the band gaps present in the non-lossy Floquet-driven system close. In this way, a chiral Floquet band is established that winds around the two-dimensional Floquet-Bloch Brillouin zone, and which thus carries quantized transport given by the Chern number. We emphasize that this is not merely due to a dissipation-induced band smearing, but a true renormalization of the real part of the energy eigenvalues, induced by the nonlinearity of the eigenvalue equation. In a real-space picture, the phenomenon of gap closing can be understood as selective suppression of one of the counter-propagating states. In order to examine the theoretical predictions, we used evanescently coupled plasmonic waveguide arrays to implement the model. Combining real-space and Fourier-space imaging, we demonstrated fast, quantized transport in the waveguide arrays. In real space, the center of mass of the excited surface-plasmon polariton wave packet was shifted by one unit cell per driving cycle. In Fourier space quantized pumping is seen as a chiral Floquet band that winds around the quasienergy Brillouin zone. Additional experiments showed that, first, unlike in a simple combination of directional couplers, the SPP transport in our system is independent on the driving frequency. Second, the transport in the opposite direction is strongly suppressed. Our experimental results agree well with the theoretical predictions based on Floquet theory.

Our findings may open a new line of research using dissipative Floquet engineering to control periodically driven quantum systems. Specifically, it will be interesting to see whether in a conserving system time-periodic imaginary parts in an effective single-particle equation of motion can be induced not only by losses but rather by interactions, and if they can be controlled so

as to establish topologically nontrivial, effective band structures. The present plasmonic waveguide setup constitutes a model for man-body systems with particle loss to an external bath. The latter are often described by the Lindblad formalism, if the bath is Markovian. The topological structure of a periodically driven system, however, becomes visible in Floquet space only. In a many-body description, this would call for a combination of the Floquet and the Lindblad techniques, which is a combination is a fundamental, unresolved problem[37]. Note added in proof: After submission of the final manuscript, a study on a very similar subject appeared[38].

## Methods

**Non-Hermitian Floquet theory.** In momentum space the Hamiltonian of the driven Rice-Mele model with periodic dissipation reads,

$$\hat{H}_k(t) = (J_1 + J_2)\cos\frac{ka_0}{2}\,\sigma_x + (J_1 - J_2)\,\sin\frac{ka_0}{2}\,\sigma_y \\ + (u_a - i\gamma_a)(1 + \sigma_z)/2 + (u_b - i\gamma_b)(1 - \sigma_z)/2, \quad (3)$$

where the coefficients have the above time dependence, $\sigma_x$, $\sigma_y$, $\sigma_z$ are the Pauli matrices acting in $(A, B)$ sublattice space, and $k$ and $a_0$ denote momentum and the lattice constant, respectively.

We now develop the Floquet formalism for non-Hermitian, periodic Hamiltonians. Due to time periodicity, the eigenstates of $\hat{H}_k$ obey the Floquet theorem[39–41],

$$|\Psi_{k\alpha}(t)\rangle = e^{-i\varepsilon_{k\alpha}t}|\phi_{k\alpha}(t)\rangle, \quad (4)$$

where a Greek index $\alpha \in \{1, 2\}$ denotes the band quantum number originating from the two sublattices, and $|\phi_{k\alpha}(t)\rangle = |\phi_{k\alpha}(t + T)\rangle$ are time-periodic states which, by construction, obey the Floquet equation

$$\mathcal{H}_k(t)|\phi_{k\alpha}(t)\rangle = \varepsilon_{k\alpha}|\phi_{k\alpha}(t)\rangle, \quad (5)$$

with $\hat{\mathcal{H}}_k(t) := [\hat{H}_k(t) - i\partial_t]$. The non-Hermiticity is accounted for by complex

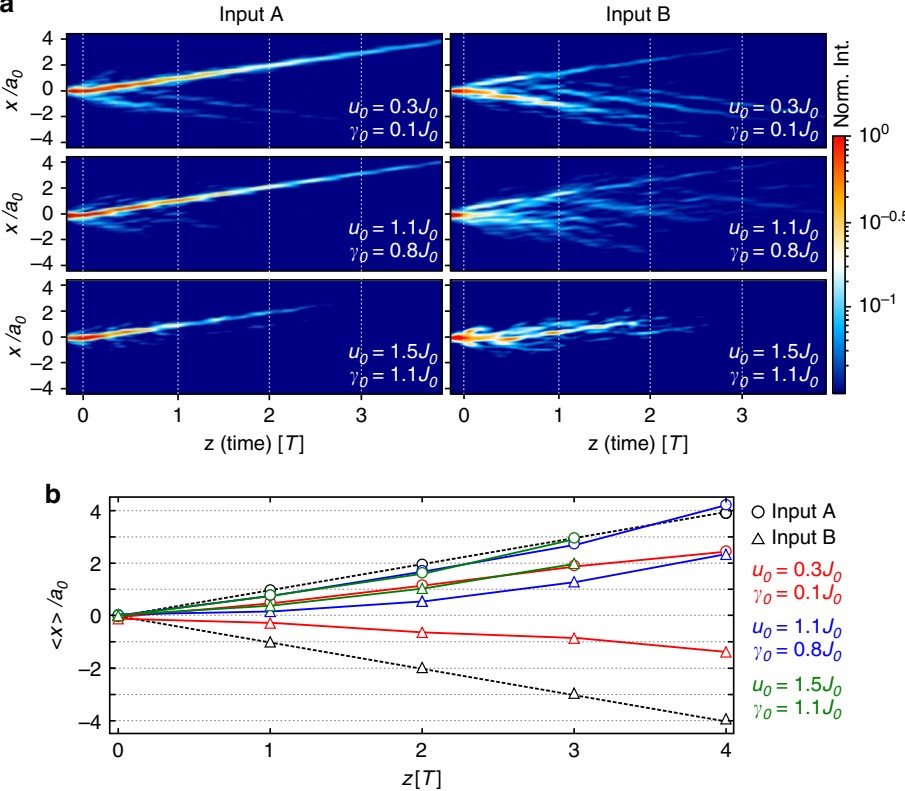

**Fig. 7 Influence of cross-section modulation and input conditions on the transport. a** Real-space SPP intensity distributions for the arrays with different strengths of the cross-section modulation ($u_0 = 0.3J_0$, $\gamma_0 = 0.1J_0$), ($u_0 = 1.1J_0$, $\gamma_0 = 0.8J_0$), and ($u_0 = 1.5J_0$, $\gamma_0 = 1.1J_0$). Measurements on the left-hand side show the SPP propagation after excitation at sublattice A (low-loss input). Measurements on the right-hand side show the SPP propagation after excitation at sublattice B (high-loss input) (**b**) The CoM position of the SPP intensity in dependence on propagation distance $z$ calculated from the experimental results shown in **a**. Note that the $z$-axis here is normalized to the period $T$. The black dashed line shows the anticipated adiabatic behaviour.

quasienergies $\varepsilon_{k\alpha}$. Note that for nonlinear or interacting, dissipative systems the Floquet theorem would generally not hold due to non-periodic, decaying density terms in the Hamiltonian. Expanding the $|\phi_{k\alpha}(t)\rangle$ in the basis of time-periodic functions, $|\phi_{k\alpha}(t)\rangle = \sum_n e^{-in\Omega t} |u_{k,\alpha}^n\rangle$ (Floquet representation), Eq. (5) takes the form of a discrete matrix Floquet–Schrödinger equation,

$$\sum_{l,\gamma} (\mathcal{H}_k)_{\beta\gamma}^{nl} \, u_{k,\gamma\alpha}^{lm} = \varepsilon_{k\alpha} \, u_{k,\beta\alpha}^{nm} \,, \tag{6}$$

where $(\mathcal{H}_k)_{\beta\gamma}^{nl} = [(H_k)_{\beta\gamma}^{nl} - n\Omega \, \delta^{nl}\delta_{\beta\gamma}]$ is the time-independent Floquet Hamiltonian and $(H_k)_{\beta\gamma}^{nl}$ the representation of $\hat{H}_k(t)$ in the basis of time-periodic functions, $\{e^{-in\Omega t} | n \in \mathbb{Z}\}$. Equation (6) determines the eigenvalues $\varepsilon_{k\alpha}$ and the eigenvector components $u_{k,\beta\alpha}^{nm} \in \mathbb{C}$ for the above Floquet expansion. Since there are as many eigenvectors as the dimension of the Floquet Hamiltonian, these components not only carry a RM sublattice index $\beta$ and a Floquet expansion index $n$, but also a band index $\alpha$ and a Floquet index $m$ to label the different eigenvectors. Thus, $(u_{k,\beta\alpha}^{nm})$ is the matrix comprised of column eigenvectors of Eq. (6). Note that Eq. (6) cannot be diagonalized separately in the sublattice space ($\beta\alpha$) and in the Floquet space ($nm$) because of the entanglement of both spaces.

In quantum mechanics, expectation values of an observable $\hat{A}$ are calculated as overlap matrix elements like $\langle\Psi|\hat{A}|\Psi\rangle$, which defines the standard scalar product in Hilbert space. However, the eigenstates of a NH Hamiltonian $\hat{H}_k$ are generally not simultaneously eigenstates of $\hat{H}_k^\dagger$[28,42]. As a consequence, they do not constitute an orthonormal basis with respect to the standard scalar product of quantum mechanics. This hampers the expansion of a quantum state $|\Psi(t)\rangle$, prepared with a given initial condition $|\Psi(t=0)\rangle$ as in the experiments, in terms of Hamiltonian eigenstates. For the sake of orthonormal basis expansions, a scalar product in Hilbert space can be defined by constructing the dual (bra) states $\langle\tilde{\Psi}_{k\alpha}|$ corresponding to the ket states $|\Psi_{k\alpha}\rangle$ in the following way. When $|u_{k\alpha}^m\rangle$ is an eigenstate of Eq. (6), it is clear that there exists an, in general different, adjoint state $|\tilde{u}_{k,\alpha}^m\rangle$ such that

$$\mathcal{H}_k^\dagger |\tilde{u}_{k,\alpha}^m\rangle = \varepsilon_{k\alpha}^* |\tilde{u}_{k,\alpha}^m\rangle. \tag{7}$$

The dual state is then obtained as $\langle\tilde{u}_{k,\alpha}^m| = |\tilde{u}_{k,\alpha}^m\rangle^\dagger$, defining the scalar product as $\langle\tilde{u}_{k,\alpha}^m|u_{k',\beta}^n\rangle$. Using Eq. (6) and the Hermitian conjugate of Eq. (7), it is easy to show that the Floquet states fulfill the biorthonormality (and corresponding completeness) relation (for non-degenerate $\varepsilon_{k\alpha} \neq \varepsilon_{k'\beta}$)

$$\langle\tilde{u}_{k\alpha}^m|u_{k'\beta}^n\rangle = \sum_{l,\gamma} (\tilde{u}_{k,\alpha\gamma}^{ml})^* \, u_{k',\gamma\beta}^{ln} = \delta_{kk'} \, \delta_{\alpha\beta} \delta^{mn} \,. \tag{8}$$

The retarded Green's function to the NH Hamiltonian $\mathcal{H}_k$ is then the causal part of the time evolution operator in Floquet representation,

$$G_{k,\beta\alpha}^{nm}(t-t') = -i\Theta(t-t') \langle\tilde{u}_{k\beta}^n| e^{-i\mathcal{H}_k(t-t')} |u_{k\alpha}^m\rangle, \tag{9}$$

which yields the spectral representation

$$G_{k,\beta\alpha}^{nm}(E) = \sum_{l,\gamma} \frac{(\tilde{u}_{k,\beta\gamma}^{nl})^* \, u_{k,\gamma\alpha}^{lm}}{E - \varepsilon_\gamma - l\Omega + i0} \,. \tag{10}$$

Note that the lossy dynamics (Im $\varepsilon_{k\alpha} \leq 0$) ensures the convergence of the Fourier integral.

An arbitrary state $|\Psi(t)\rangle$ can now be expanded in the basis of Floquet states as

$$|\Psi(t)\rangle = \sum_{k,\alpha,n} C_{k\alpha}^n e^{-i(\varepsilon_{k\alpha}+n\Omega)t} |u_{k,\alpha}^n\rangle, \quad C_{k\alpha}^n = \langle\tilde{u}_{k,\alpha}^n|\Psi(0)\rangle, \tag{11}$$

where the time-independent expansion coefficients $C_{k\alpha}^n$ are calculated at the initial time $t = 0$ using the biorthonormality relation (Eq. (8)) and, thus, incorporate the initial conditions on $|\Psi(t)\rangle$.

Using the expansion (Eq. (11)), physical expectation values for time-evolving states can now be calculated in a straight-forward way and decay exponentially in time due to the lossy dynamics of the system. For instance, the density of a driven-dissipative Floquet state reads,

$$\langle\Psi_{k\alpha}(t)|\Psi_{k\alpha}(t)\rangle = e^{-\Gamma_{k\alpha}t}, \tag{12}$$

with the decay rate $\Gamma_{k\alpha} = -2\text{Im}\varepsilon_{k\alpha} > 0$. In our DLSPPW experiments below it is possible to directly measure the momentum-resolved and energy-resolved

population density, i.e., intensity of the Fourier transform $|\Psi_k(E)\rangle$, which reads,

$$I(E,k) = \langle \Psi_k(E) | \Psi_k(E) \rangle$$

$$= \sum_{n,m,\alpha\beta} \sum_{l,\gamma} \frac{C_{k\beta}^{l*} C_{k\alpha}^{l} \left(u_{k\beta\gamma}^{nl}\right)^* u_{k,y\alpha}^{lm}}{(E - \varepsilon_{k\beta}^* - l\Omega - i0)(E - \varepsilon_{k\alpha} - l\Omega + i0)}. \quad (13)$$

It is seen that, in general, this expression involves the mixing of the RM bands $(\alpha, \beta)$, leading to a broad spectral distribution in the FBBZ. A distribution of this type is shown in Fig. 2c. It is also possible to effectively populate only one RM band $\alpha$ by populating at the initial time $T = 0$ one single site of the initially nonlossy sublattice, see Fig. 2a and b. In this case, the $\alpha - \beta$ cross terms vanish, and Eq. (13) simplifies to

$$I_\alpha(E,k) = \langle \Psi_{k\alpha}^n(E) | \Psi_{k\alpha}^n(E) \rangle$$

$$= \sum_{n,m,l,\gamma} C_{k\alpha}^{l*} C_{k\alpha}^{l} \frac{\left(u_{k\alpha\gamma}^{nl}\right)^* u_{k,y\alpha}^{lm}}{|E - \varepsilon_{k\alpha} - l\Omega|^2}. \quad (14)$$

This is similar, albeit not identical, to the spectral density obtained from the imaginary part of the Green's function in Eq. (9). Thus, measurements of the population density $I(E, k)$ of a wave function initialized at $t = 0$ provide detailed information about the stationary spectral function.

**Dissipative transport quantization.** For an adiabatic Thouless pump the number of particles transported by one lattice constant per cycle is given by the Berry phase, i.e., the Berry flux penetrating a closed loop in Hamiltonian parameter space. Therefore, it is quantized and time plays no role[1]. For fast driving, considering the states localized on single sites as approximate eigenstates for small hopping amplitude, the driving-induced hopping to neighbouring sites can be viewed as Landau-Zener tunnelling in real space[29]. However, the topological nature of the process is better analyzed by working in momentum and frequency space. Namely, any nonzero driving frequency $\Omega$ turns the problem into an effectively two-dimensional (2D) one due to the periodicity in space and time. In this case, the Hermitian RM model possesses two counterpropagating chiral Floquet bands in the 2D FBBZ $\{-\Omega/2 \le \varepsilon < \Omega/2; -\pi/a_0 \le k < \pi/a_0\}$[15,43], as depicted in Fig. 2a. Quantized transport in a Floquet band is controlled by the winding or Chern number of the band around the FBBZ[17]. Here we investigate transport quantization in a general, fast pumped, dissipative situation. The velocity operator reads $\hat{v} = \mathrm{Re}\, d\hat{\mathcal{H}}_k/dk$ ($h = 1$)[17], i.e., for each $k$-state $|\Psi_{k\alpha}(t)\rangle$, its eigenvalue is the group velocity $d\mathrm{Re}\, \varepsilon_{k\alpha}/dk$. Thus, the spatial displacement of the particle number during one pumping cycle carried by a single Floquet state $|\Psi_{k\alpha}(t)\rangle$ with a loss rate $\Gamma_{k\alpha}$ is given by

$$\int_0^T dt \frac{\langle \Psi_{k\alpha}(t) | \hat{v} | \Psi_{k\alpha}(t) \rangle}{\langle \Psi_{k\alpha}(t) | \Psi_{k\alpha}(t) \rangle} = \frac{d\mathrm{Re}\, \varepsilon_{k\alpha}}{dk} T. \quad (15)$$

Note that the velocity expectation value is normalized by the exponentially decaying probability density, Eq. (12), such that in Eq. (15) the exponential decay factor $\exp(-\Gamma_{k\alpha}t)$ drops out. The shift per cycle carried by a band $\alpha$ with population density $I_\alpha(E, k)$ (c.f. Eq. (13)) is obtained by integrating over the FBBZ, that is, over the energy $E$ and all $k$ states, and reads,

$$L_\alpha = \int_{\mathrm{FBBZ}} \frac{dE}{\Omega} \int_{-\pi/a_0}^{\pi/a_0} \frac{dk}{2\pi/a_0}\, I_\alpha(E,k) \frac{d\mathrm{Re}\, \varepsilon_{k\alpha}}{dk} T. \quad (16)$$

For a homogenously filled band, $\int dE\, I_\alpha(E, k)/\Omega = 1$, this reduces to

$$\frac{L_\alpha}{a_0} = \int_{-\pi/a_0}^{\pi/a_0} \frac{dk}{2\pi} \frac{d\mathrm{Re}\, \varepsilon_{k\alpha}}{dk} T = Z. \quad (17)$$

For a periodically driven system, the dispersion $\mathrm{Re}\, \varepsilon_{k\alpha}$ is not only a periodic function of $k$, but its values are also periodic with period $\Omega$. That is, $\varepsilon_{k\alpha}$ is a mapping from the 1D circle onto the 1D circle, $\mathrm{Re}\, \varepsilon_\alpha : S^1 \to S^1$, and wraps around the 2D torus of the FBBZ as shown in Fig. 2d. Equation (17) is the definition of the winding number around the circle. It is seen that it assumes nonzero, integer values if the dispersion continuously covers the entire FBBZ in the frequency direction, i.e., if it is gapless, $\int dk\, d\mathrm{Re}\, \varepsilon_{k\alpha}/dk = Z\Omega = Z \frac{2\pi}{T}$, since $\varepsilon_{\pi/a_0,\alpha} = \varepsilon_{-\pi/a_0,\alpha}$. This proves the second equality in Eq. (17) for a gapless dispersion and indicates transport quantization.

**Samples.** The DLSPPW arrays are fabricated by negative-tone gray-scale electron beam lithography (EBL)[30,44]. The waveguides consist of poly(methyl methacrylate) (PMMA) ridges deposited on top of a 60 nm thick gold film evaporated on a glass substrate. The mean center-to center distance between the ridges is 1.7 μm and the maximum deflection from the center is 0.5μm. The resulting variation of coupling constants is $J_1(z) = J_0 e^{-\lambda(1-\sin\Omega z)}$, $J_2(z); = J_1(z - T/2)$ with $J_0 = 0.144$ μm$^{-1}$ and $\lambda = 1.75$.

The cross-section of each waveguide is controlled by the applied electron dose during the lithographic process. By varying the electron dose along the z-axis we modulate the waveguides' cross-sections and hence the propagation constants as $\beta_a(z) \approx \bar{\beta} - u_0 \cos(\Omega z + \varphi) - i\gamma_a(z)$, $\beta_b(z) = \beta_a(z - T/2)$, where $\bar{\beta} =$

$6.62 + i0.015$ μm$^{-1}$ corresponds to the mean height 100 nm and the mean width 250 nm of a waveguide and $\gamma_a(t) \approx -\gamma_0 \Theta(u_a(z)) \cos(\Omega z + \varphi)$ is the periodic loss rate induced by coupling to free SPPs. The choice of such geometrical parameters is motivated by the fact that strong losses due to coupling to continuum of free propagating SPPs occur when the height and the width of a waveguide are smaller than the corresponding mean values, i.e., $\beta_j(z) < \bar{\beta}$. Other sources of losses can be assumed to be independent of $z$ because their variation is negligibly small in comparison to this effect.

**Leakage radiation microscopy.** SPPs are excited by focusing a TM-polarized laser beam with free space wavelength $\lambda_0 = 980$ nm (NA of the focusing objective is 0.4) onto the grating coupler deposited on top of the central waveguide (either sublattice A or B). The propagation of SPPs in an array is monitored by real-space and Fourier-space leakage radiation microscopy[45,46]. For this purpose, we use an oil immersion objective (×63 magnification, NA = 1.4) to collect the leakage radiation. Real space intensity distributions are recorded by imaging the sample plane onto a CMOS camera. The corresponding Fourier images are obtained by imaging the back-focal plane of the objective onto the camera. The directly transmitted laser beam is blocked by Fourier filtering. We note that we work in the single-mode waveguide regime for all cross sections used in the experiments at the design wavelength.

## Data availability
The data that support the findings of this study are available from the corresponding author upon reasonable request. All these data are directly shown in the corresponding figures without further processing.

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

## Acknowledgements

This work was supported by the Deutsche Forschungsgemeinschaft (DFG) within SFB/TR 185 (277625399) and the Cluster of Excellence ML4Q (390534769).

## Author contributions

Z.F. fabricated the samples, conducted the experiments, and performed the numerical calculations with the help of H.Q. S.L. and J.K. conceived the project and supervised Z.F. and H.Q., respectively. All authors contributed to the discussion and interpretation of the results, as well as writing the manuscript.

## Competing interests

The authors declare no competing interests.
