## [Peer Review File · Nature Communications]

REVIEWER COMMENTS

Reviewer #1 (Remarks to the Author):

This manuscript deals with non-Hermitian Floquet physics, more precisely with a theoretical analysis and experimental realisation of time-periodic dissipation-induced enhancing of Thouless pumping quantization in the driven Rice-Mele model.

The experimental realisation consists of arrays of coupled waveguides, with the propagation direction z playing the role of time, in the paraxial approximation to the Helmholtz equation.

The theoretical analysis introduces the concept of time-periodic modulation of dissipation, and explores the consequences in a non-Hermitian Floquet framework.

The combination of theory and experiment, both careful and well described, realised in this manuscript is quite convincing, and I would be in favour of publication.

There is one aspect of the discussion that I think the authors should improve. One of the common approaches to open quantum systems, especially in the context of quantum optics, goes through a Markovian quantum master equation of the Lindblad form. There, the non-Hermitian terms are supplemented by quantum jump terms which, conspiring together, allow, for instance, reaching stationary solutions in time-independent cases.

I now understand that such a Lindblad approach belongs in some way to a 'different community' of physicists. Still, I would love to read, in the present manuscript, a discussion of the connections of the present non-Hermitian dissipation approach to the other, Lindbladian, framework. (Incidentally, the Lindblad approach is much less developed in the explicitly time-dependent case, which would be relevant here.)

Minor corrections/typos:

Schrödinger in place of Schrödinger on page 1

quantization in place of quantization in the caption of Figure 2.

cite in place of site on pag. 4.

In the text description of Fig. 7 (b) it is said '(waveguide A: circles, waveguide B: squares)', but I see no squares; rather triangles.

Reviewer #2 (Remarks to the Author):

In this work, the authors construct a dissipative variant of Rice-Mele topological pumping that leads to quantized center-of-mass displacement in periodically-modulated coupled polariton waveguide array. The work is of contemporary interest when topological effects are explored both in time-dependent systems, as well as in non-Hermitian systems. The extension of topological pumping and its realization in a finite time period alongside dissipation, highlights an important distinction to adiabatic equilibrium idealizations. In this sense, I find the work relevant for publication in Nat. Commun. given that the authors can address/answer the following points:

1. The authors heavily rely on the fact that at their input facet, the Wannier orbitals of each band of the Rice-Mele model are localized on a single site. Thus, they can excite effectively a single band. Had the initial state in the cycle corresponded to another configuration along the pump cycle, this would not have worked. I would ask the authors to highlight this point in the paper.
2. Similar ideas where topological bands are populated by dissipation have been proposed in the past, for example, in [Phys. Rev. Lett. 112, 133902 (2014)]. Please refer and discuss the connection to such works.
3. Can the authors discuss the FBBZ SPP also for Figs. 6 and 7.
4. Can the authors derive an analytic expression for the gap size G (to lowest orders in the Floquet-Bloch truncation - this should not be complicated) - both for the dissipative and non-dissipative case? This would corroborate the numerical analysis and will provide a clear relation between Ω and γ . Correspondingly, in Fig. 2f, why does the gap close at $\sim 0.8 \Omega$?
5. In Fig. 3b, the wavepacket excited with site B is scattering back to the other band and moves in the same direction as the one excited with site A. What kind of backscattering process leads to this behavior?
6. The Rice-Mele topological pumping is closely related to the adiabatic limit of a Landau-Zener process. The setting employed in this work is even more closely related to the dissipative version of Landau-Zener. I would like the authors to comment on this relation, and it could assist them also in answering the point 5 above.
7. In the methods section above Eq. (14), why is the constant dissipation assumption required?
8. I am confused by the quantized transport predicted by equations (14) and (15): commonly the topological pump is related to a time-dependent scan over the polarization of the populated band. It is obtained similarly to (14), where two contributions arise corresponding to a group velocity term [as appearing on the right hand side of (14)], and another geometric term that leads to the topological winding (quantized displacement). How does this comply with the presented analysis? Is the transport quantized due to topology at all?
9. Last, the center of mass displacement appears to be relatively quantized. However, the losses lead to much reduced intensity. This means that quantized charge transport will be lost. I would ask the authors to discuss this point, its prospective technological implications, and its relation to previous attempts to obtain a high-precision metrological estimation of the Ampere, see, for example [<https://journals.aps.org/prl/abstract/10.1103/PhysRevLett.101.066801>]

Reviewer #3 (Remarks to the Author):

In this paper, the authors make significant experimental and theoretical progress towards non-adiabatic charge pumping in the presence of loss. Theoretically, they make what I believe is the first prediction of quantized non-adiabatic charge pumping in non-Hermitian systems. They then confirm these predictions experimentally using dielectric-loaded surface plasmon-polariton waveguides, which enable analogues of Schrodinger equations similar to pioneering work on Floquet Chern insulators, but with the presence of controllable "time-dependent" loss. In short, I find the work quite interesting, well-written, and novel, and recommend publication in Nature Communications. I have some notes

below that should be considered to improve the paper, but are not necessary for publication.

On the theory front, I would appreciate more details of the proof of Eq. 14 and 15. As currently written, I take this as an intuitive approximate result, but mathematically there are a number of steps missing. For instance, the paper jumps back and forth between the Floquet extended zone treatment and the quasienergies. An important example of this is the statement that the norm of the modes decays as $\exp(-\Gamma_{\mathbf{k}} t)$. This would only be true in the absence of micromotion, or its non-Hermitian equivalent, whereas in practice it will definitely be the case that some points in time are lossier than others. It is not clear how this enters into the proof that charge pumping corresponds to energy winding. Without a more detailed proof, I think that Eqs. 14 and 15 should be argued as approximate statements neglecting k -dependence and t -dependence of the imaginary part of the energy, which is consistent with numerics + experiments.

Regarding the experiments, I am a bit confused by the center of mass plots, which seem to consistently show an upturn above linear behavior in most of the charge pumping data. Is this understood? If so, a few words about why this happens and why we expect it to plateau to a quantized slope for larger number of "Floquet cycles" is merited.

Response to the Reviewers' Comments

We sincerely thank all the reviewers for their positive assessment of our work as well as its presentation in the manuscript. Especially, we appreciate their constructive comments which are pointed towards the improvement of the quality of our manuscript. We read all comments carefully and answer them point-by-point below. For better readability we printed the original reviewer comments in blue, the corresponding answers in black, and the changes to the manuscript in red.

Reviewer #1

Reviewer: This manuscript deals with non-Hermitian Floquet physics, more precisely with a theoretical analysis and experimental realisation of time-periodic dissipation-induced enhancing of Thouless pumping quantization in the driven Rice-Mele model. The experimental realisation consists of arrays of coupled waveguides, with the propagation direction z playing the role of time, in the paraxial approximation to the Helmholtz equation. The theoretical analysis introduces the concept of time-periodic modulation of dissipation, and explores the consequences in a non-Hermitian Floquet framework. The combination of theory and experiment, both careful and well described, realised in this manuscript is quite convincing, and I would be in favour of publication.

There is one aspect of the discussion that I think the authors should improve. One of the common approaches to open quantum systems, especially in the context of quantum optics, goes through a Markovian quantum master equation of the Lindblad form. There, the non-Hermitian terms are supplemented by quantum jump terms which, conspiring together, allow, for instance, reaching stationary solutions in time-independent cases.

I now understand that such a Lindblad approach belongs in some way to a 'different community' of physicists. Still, I would love to read, in the present manuscript, a discussion of the connections of the present non-Hermitian dissipation approach to the other, Lindbladian, framework. (Incidentally, the Lindblad approach is much less developed in the explicitly time-dependent case, which would be relevant here.)

Response: We have, in fact, considered the Lindblad formalism as well. The lossy dynamics of waveguide arrays corresponds to particle losses in many-body systems. Particle losses to an external bath are usually described by the Lindblad formalism (Lindblad equation for the density matrix), if the bath is Markovian. The topological structure of a periodically driven system, however, becomes visible in Floquet space only. This calls for a combination of the Floquet and the Lindblad techniques. However, such a combination is a fundamental, unresolved problem. The difficulty is that even though the Hamiltonian and loss rates are time-periodic, the density matrix is NOT (states comprising the density matrix decay). Hence, an exact Floquet Fourier representation of the Lindblad equation of motion cannot easily be derived. The development of a suitable perturbative Lindblad approach with respect to the loss rates would be definitely very interesting. However, it is clearly beyond the scope of this work and we only added a brief note to the conclusion of our paper:

"The present plasmonic waveguide setup constitutes a model for many-body systems with particle loss to an external bath. The latter are often described by the Lindblad

formalism, if the bath is Markovian. The topological structure of a periodically driven system, however, becomes visible in Floquet space only. In a many-body description, this would call for a combination of the Floquet and the Lindblad techniques, which is a fundamental, unresolved problem [37].”

Reviewer: *Minor corrections/typos:*

Schröderinger in place of Schrödinger on page 1

qunatization in place of quantization in the caption of Figure 2.

cite in place of site on pag. 4.

In the text description of Fig. 7 (b) it is said `(waveguide A: circles, waveguide B: squares)', but I see no squares; rather triangles.

Response: We thank the reviewer for the careful reading and corrected the typing errors.

Reviewer #2

Reviewer: *In this work, the authors construct a dissipative variant of Rice-Mele topological pumping that leads to quantized center-of-mass displacement in periodically-modulated coupled polariton waveguide array. The work is of contemporary interest when topological effects are explored both in time-dependent systems, as well as in non-Hermitian systems. The extension of topological pumping and its realization in a finite time period alongside dissipation, highlights an important distinction to adiabatic equilibrium idealizations. In this sense, I find the work relevant for publication in Nat. Commun. given that the authors can address/answer the following points:*

1. The authors heavily rely on the fact that at their input facet, the Wannier orbitals of each band of the Rice-Mele model are localized on a single site. Thus, they can excite effectively a single band. Had the initial state in the cycle corresponded to another configuration along the pump cycle, this would not have worked. I would ask the authors to highlight this point in the paper.

Response: We thank the reviewer for pointing us towards this. We agree that indeed the single band excitation in our case stems from two conditions: input on a single sublattice and the starting point of the parametric cycle, i.e. the “phase” of the driving force. The latter factor was not highlighted in our manuscript. In order to address this point, we have added the following sentence to page 2:

“For the parametric cycle configuration shown in Fig. 1 (b) the chosen initial time moment leads to an asymmetric amplitude distribution of the counter-propagating Floquet states with respect to the two sublattices. It is seen in Fig. 2 that such initial conditions populate, by Fourier expansion, almost homogeneously an entire right- or left-moving band.”

Reviewer: 2. Similar ideas where topological bands are populated by dissipation have been proposed in the past, for example, in [Phys. Rev. Lett. 112, 133902 (2014)]. Please refer and discuss the connection to such works.

Response: We thank the reviewer for this remark. In this paper (new reference 26 in the revised manuscript) the authors proposed to use dissipation to measure topological quantities in photonic lattices. Another example of probing topological invariants with losses was described earlier in Phys. Rev. Let. 102, 065703 (2009) (new reference 25 in the revised manuscript). In contrast to these works, we employ dissipation to restore topological properties of our system. Another important distinction is that we modulate the losses in time which allows us to affect the Floquet bands, while in the mentioned papers the authors consider constant decay rate. In order to complete the discussion of the effects of non-Hermiticity in the introduction part, we have included these references and added the following sentence to page 1:

“Non-Hermiticity has been utilized to probe topological quantities [25,26]”.

Reviewer: 3. Can the authors discuss the FBBZ SPP also for Figs. 6 and 7.

Response: As the reviewer proposes, we have added to the manuscript the corresponding discussions:

For Fig.6: “In Fourier space changing the modulation frequency influences primarily the width of the Floquet BZ: the lower is the frequency, the smaller is the distance between the neighboring bands and the smaller is the tilt of these bands which reflects the wavepacket group velocity in absolute values (data not shown).”

For Fig. 7: “In Fourier space increasing the modulation strength results in a strong band broadening caused by a growing damping rate. This effect is more pronounced for the input B (data not shown).”

To illustrate our discussion, we show below the corresponding momentum-resolved spectra. If the reviewer considers it useful, we will add these images to the final version of the manuscript as supplementary material.

Figure 1 Fourier-space data corresponding to Fig. 6 (a). The images on the left-hand side correspond to the case with cross-section modulation. As the frequency increases the Floquet bands get closer (see arrows) and flatter, nevertheless they remain continuous which is a hallmark of quantized displacement in Real space. On the right-hand side the data for constant cross-section modulation is displayed. Here, changing the frequency has an additional effect on the band

structure: the gaps at the FBZ boarder start to open (see green circles). Such a behavior of the quasienergies results in the decreasing CoM shift as confirmed by Fig.6 (b).

Figure 1 Fourier-space data corresponding to Fig. 7 (a). The increasing modulation strength has two main effects on the momentum-resolved spectrum: first, the bands get broadened due to higher losses, second, the difference between inputs A and B gets more and more pronounced. In case of the input B the broadening effect is much stronger because of the populated high-loss band.

Reviewer: 4. *Can the authors derive an analytic expression for the gap size G (to lowest orders in the Floquet-Bloch truncation - this should not be complicated) - both for the dissipative and non-dissipative case? This would corroborate the numerical analysis and will provide a clear relation between Ω and γ . Correspondingly, in Fig. 2f, why does the gap close at $\sim 0.8 \Omega$?*

Response: An analytic expression for the gap size would certainly be much desirable. For a periodic potential, the gap size G is given to lowest order perturbation theory (which corresponds to the lowest-order truncation of the Floquet-Bloch matrix) by two times the leading Fourier component of the periodic potential. In the present case, the potential is periodic in time. Therefore, the gap opens at the Brillouin zone boundary along the frequency direction (y axis in Fig. 2 of the manuscript). However, we find quantitatively that, for the large oscillation amplitudes considered here, this perturbative estimate does not agree well with the exact, numerical result, which means that the higher Floquet bands have a large influence on the gap size and a low-order truncation is not valid. For instance, for the parameter values specified in the chapter “Results”, at the beginning of section “Non-Hermitian Floquet analysis”, perturbation theory would give the gap size of roughly $G=4J_0$, much larger even than the width of the FBBZ of $\Omega=1.1J_0$ (!), while the actual gap size is about 0.1 of the FBBZ width, as seen in Fig. 2 of the manuscript. We have, therefore, decided not to discuss the perturbative result in the manuscript. In case the reviewer would find it useful, we would include the above discussion in the manuscript.

Reviewer: 5. *In Fig. 3b, the wavepacket excited with site B is scattering back to the other band and moves in the same direction as the one excited with site A. What kind of backscattering process leads to this behavior?*

Response: The reversal of propagation direction for initial excitation of a B site is a consequence of the fact that the phase relations between the oscillating nearest-neighbor hopping amplitudes J_1 , J_2 and the oscillating loss amplitude do not allow an efficient population of the backward propagating chiral band. Microscopically speaking: Exciting the B site populates predominantly the backward-propagating chiral band, while exciting the A

site populates the forward propagating band. Thus, when initially the B site is excited, the excitation moves backward during the following cycle. However, the phase relation of the loss amplitude is such that at the beginning of the cycle the losses are large so that this backward transported intensity is suppressed. During this cycle, some of the intensity also spreads to the A site, thus populating the forward propagating band, and, therefore, moves forward during the subsequent cycles, as seen in Fig. 3 b. Thus, the reversal of propagation direction is a signature of the chirality of the bands, rather than a scattering process. This effect was briefly explained in the last sentence before the chapter “Experiments”. We have expanded this sentence to clarify the explanation:

“This is a signature of the chirality of the Floquet bands and is due to the fact that the propagation of even poorly populated low-loss states in positive x -direction starts to dominate after the first few periods, while the states propagating in negative x -direction are quickly damped due to the phase relation of the periodic losses with respect to the hopping amplitude.”

Reviewer: 6. The Rice-Mele topological pumping is closely related to the adiabatic limit of a Landau-Zener process. The setting employed in this work is even more closely related to the dissipative version of Landau-Zener. I would like the authors to comment on this relation, and it could assist them also in answering the point 5 above.

Response:

When a system’s state is driven nonadiabatically through an avoided crossing of two bands, a transition from one band to the other can occur with the probability given by the famous Landau-Zener formula. For the present system, considering the localized states on single sites as approximate eigenstates of the system, the hopping between neighboring sites induced by the fast periodic driving can be viewed as Landau-Zener tunneling in real space. For this reason, Landau-Zener formula is often used as an adiabaticity criteria for the Rice-Mele model (see Ref. 13). In case of the dissipative version of Landau-Zener, the non-adiabatic transitions can be eliminated even for arbitrary fast processes as shown for instance in Ref. 29. As fairly pointed by the reviewer, it is closely related to the observed non-adiabatic quantized transport in our work. However, to analyze the topological structure, it is better to work in momentum and Floquet frequency space. Namely, for fast Floquet driving, the band structure and, hence, the avoided crossing (gap) is generated by the Floquet driving itself, and the Floquet Hamiltonian, Eq. (6), which exhibits this band structure, is time independent. The analogy to Landau-Zener could be confusing for the chosen Floquet representation as the fast driving does not induce transitions across the gap. Rather the closing of the gap is purely due to the dissipative dynamics and the resulting modification of the band structure, as described in the paper.

In order to discuss this relation of Landau-Zener tunneling to the case of fast Floquet driving, we have added the following two sentences at the beginning of the section “Dissipative transport quantization”, and we also cite Ref. [29] in this respect, where non-Hermitian Landau-Zener tunneling is discussed for a non-topological system:

“For fast driving, considering the states localized on single sites as approximate eigenstates for small hopping amplitude, the driving-induced hopping to neighboring sites can be viewed as Landau-Zener tunneling in real space [29]. However, the topological

nature of the process is better analyzed by working in momentum and frequency space. Namely, ...”

Reviewer: 7. *In the methods section above Eq. (14), why is the constant dissipation assumption required?*

Response: This assumption is actually not required, and the dissipation parameter $\Gamma_{k\alpha}$ drops out of the calculation. We thank the referee for pointing this out. We have removed the corresponding sentence from the manuscript before Eq. (14) (now Eq. (15) in the new version) and have expanded the derivation around Eq. (15), also in response to the Reviewer’s point 8 and to Reviewer #3 (see below), and discuss there explicitly why the dissipation drops out.

Reviewer: 8. *I am confused by the quantized transport predicted by equations (14) and (15): commonly the topological pump is related to a time-dependent scan over the polarization of the populated band. It is obtained similarly to (14), where two contributions arise corresponding to a group velocity term [as appearing on the right hand side of (14)], and another geometric term that leads to the topological winding (quantized displacement). How does this comply with the presented analysis? Is the transport quantized due to topology at all?*

Response: (The equation numbers refer to the numbering in the new version of the manuscript.) Indeed, for adiabatic pumping the transported charge is given by the quantized Berry phase (or Chern number) that arises from a closed curve in parameter space. In the present case of fast, nonadiabatic pumping, however, the topological map is different, namely the dispersion which is a mapping from the 1D circle (k-space) onto the 1D circle (frequency with Floquet period Ω). In this case, the quantized CoM displacement is given by the winding number of the band dispersion around the FBBZ torus along the frequency axis. This winding number is integer and nonzero, precisely if the dispersion is gapless. We have considerably expanded the derivation and explaining text around Eqs. (15)-(17) to clarify this fact. We thank the referee for bringing up this question which helped us to clarify the topological character of the fast, nonadiabatic pumping.

Reviewer: 9. *Last, the center of mass displacement appears to be relatively quantized. However, the losses lead to much reduced intensity. This means that quantized charge transport will be lost. I would ask the authors to discuss this point, its prospective technological implications, and its relation to previous attempts to obtain a high-precision metrological estimation of the Ampere, see, for example*

[<https://journals.aps.org/prl/abstract/10.1103/PhysRevLett.101.066801>]

Response: The reviewer is completely right, the center-of-mass displacement is quantized, but the transported charge is not, due to the lossy dynamics of the system. Previous attempts to introduce a current standard (e.g. the reference cited by the reviewer) used single-electron transistors (SET), where the charge quantization is not topologically stabilized but by a large Coulomb blockade, and is therefore approximate. Our present work is of fundamental character and does not directly aim at introducing a metrological current standard. It rather opens up the path how non-Hermitian Schrödinger dynamics can induce transport quantization, and that this behavior can be realized in plasmonic waveguide arrays. We are, however, also thinking about charge quantized transport, not only quantized displacement: In periodically driven, many-body electronic

systems, non-Hermitian, periodic terms can be induced in the effective single-particle Hamiltonian by two-body interactions via the imaginary part of the selfenergy. Such a system would be effectively non-Hermitian, but particle-number conserving, and could thus provide a basis for a metrological current standard. In the manuscript, this possibility has already been described as an outlook at the end of the chapter “Discussion”. We find it too early to discuss a current standard in the present paper beyond this outlook remark, but we will do so if the reviewer would find it useful. In fact, we are pursuing research along the line of interaction-induced, effective non-hermiticity within a different project, which is outside the scope of the present work.

Reviewer #3

In this paper, the authors make significant experimental and theoretical progress towards non-adiabatic charge pumping in the presence of loss. Theoretically, they make what I believe is the first prediction of quantized non-adiabatic charge pumping in non-Hermitian systems. They then confirm these predictions experimentally using dielectric-loaded surface plasmon-polariton waveguides, which enable analogues of Schrodinger equations similar to pioneering work on Floquet Chern insulators, but with the presence of controllable “time-dependent” loss. In short, I find the work quite interesting, well-written, and novel, and recommend publication in Nature Communications. I have some notes below that should be considered to improve the paper, but are not necessary for publication.

On the theory front, I would appreciate more details of the proof of Eq. 14 and 15. As currently written, I take this as an intuitive approximate result, but mathematically there are a number of steps missing. For instance, the paper jumps back and forth between the Floquet extended zone treatment and the quasienergies. An important example of this is the statement that the norm of the modes decays as $\exp(-\Gamma_{k\alpha} t)$. This would only be true in the absence of micromotion, or its non-Hermitian equivalent, whereas in practice it will definitely be the case that some points in time are lossier than others. It is not clear how this enters into the proof that charge pumping corresponds to energy winding. Without a more detailed proof, I think that Eqs. 14 and 15 should be argued as approximate statements neglecting k -dependence and t -dependence of the imaginary part of the energy, which is consistent with numerics + experiments.

Response: (The equation numbers refer to the numbering in the new version of the manuscript) We have substantially expanded the derivation and explaining text around Eq. (15), showing all the steps that lead to the expression (15) for the displacement in each Floquet mode and the expression (17) for the total, quantized displacement. We have also typeset Eq. (12) as a display equation instead of an in-line equation, so that we can explicitly refer to it in the subsequent text. Thus, the quantized CoM transport is not merely an intuitive approximation, but a rigorous result.

Regarding the reviewer’s specific example of the temporal decay, he/she is right that the system is lossier in some points of time than in others. However, it is a rigorous result of Floquet theory that any Floquet wave function has the form (Eq. (4)),

$$|\Psi_{k\alpha}(t)\rangle = e^{-i\varepsilon_{k\alpha}t}|\phi_{k\alpha}(t)\rangle .$$

Here, the exponential factor, with complex quasienergy $\epsilon_{k\alpha}$, accounts for the overall decay with the time-independent rate $\Gamma_{k\alpha}$, while the time-periodic state $|\Phi(t)\rangle$ will in general have a time-dependent, but periodic amplitude (micromotion). However, in the ratio of the expectation values involved in Eq. (15) both of these factors drop out, so that the quantized displacement per cycle does not depend on micromotion or time-dependent modulation of the losses.

Reviewer: Regarding the experiments, I am a bit confused by the center of mass plots, which seem to consistently show an upturn above linear behavior in most of the charge pumping data. Is this understood? If so, a few words about why this happens and why we expect it to plateau to a quantized slope for larger number of “Floquet cycles” is merited.

Response: In the beginning of propagation, slightly lower propagation speed is expected due to non-ideal excitation conditions in the experiment. When guided SPPs are excited by shining laser onto the grating coupler, the laser spot slightly excites the neighboring waveguides and free propagating SPPs which exist at the interface between air and gold. These factors inevitably contribute to the CoM displacement. After the first cycle, these contributions have essentially died out and the CoM displacement assumes the quantized values (unit slope of the CoM plots). For the very last points (at large z) the signal intensity has decayed so far that the measurement errors of the CoM position (camera noise) increase. This explains a statistical deviation from the unit slope for large distances. In order to address this point, we have added the following sentence to the manuscript (page 5):

“We note that the somewhat lower than unit slope of the CoM plots in Figs. 6 (b), 7 (b) during the first pumping cycle (see Fig. 6) is an artifact which arises from non-ideal excitation conditions, such as weak excitation of the neighboring waveguides. The deviations at large distance are statistical and result from increasing measurement errors due to camera noise and decaying signal intensity.”

REVIEWERS' COMMENTS:

Reviewer #1 (Remarks to the Author):

I am completely satisfied with the changes made to the manuscript. I believe that the manuscript can be accepted in the present form.

Reviewer #2 (Remarks to the Author):

I thank the authors for the detailed reply. In their reply, they leave the decision re adding a comment about the nonperturbative nature of the Floquet gap. I would have added a comment about it in the paper - otherwise, other readers would be similarly confused about this point.

Otherwise, I believe that this very nice result is now ready for publication.

Reviewer #3 (Remarks to the Author):

Looks good. Thank you for addressing my comments.

Response to the Reviewers' Comments

We are happy that we have managed to successfully address all the questions raised by the reviewers and we would like to thank them again for their work, which has substantially improved the quality of our manuscript.

Reviewer #1

Reviewer: I am completely satisfied with the changes made to the manuscript. I believe that the manuscript can be accepted in the present form.

Reviewer #2

Reviewer: I thank the authors for the detailed reply. In their reply, they leave the decision re adding a comment about the nonperturbative nature of the Floquet gap. I would have added a comment about it in the paper - otherwise, other readers would be similarly confused about this point.

Otherwise, I believe that this very nice result is now ready for publication.

Response: We have added the following sentences to the section "Non-Hermitian Floquet analysis":

"Note that computing the gap size G , as visible in Fig. 2 (a), involves diagonalization of the entire Floquet Hamiltonian matrix. In leading order perturbation theory, G would be given by the Fourier amplitude of the periodic drive, i.e., for the first FBBZ by J_0 which strongly differs from the exact value."

Reviewer #3

Looks good. Thank you for addressing my comments.